# Do You Want Sustainable Olympics? Environment, Disaster, Gender, and the 2020 Tokyo Olympics

**Eiji Yamamura** 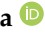

Department of Economics, Seinan Gakuin University, Fukuoka 814-8511, Japan; yamaei@seinan-gu.ac.jp

**Abstract:** The slogans of the 2020 Tokyo Olympics were "symbol of resilience from the Great East Japan Earthquake" and "Compact Olympics". The Olympics were also expected to demonstrate "gender equality" and to enhance sustainability in modern society. However, in practice, the cost of the Tokyo Olympics 2020 was far greater than estimated. The slogan was changed to "symbol of overcoming COVID-19" although in reality, infection spread dramatically during the games. Overall, the 2020 Tokyo Olympics did not turn out as expected or meet the expectations of the populace. Using individual-level data, we tested how and to what extent Japanese proponents of a sustainable society supported the compact Olympics announced in 2016. The key findings are: (1) most people support policies for environmental protection, gender equality, and disaster prevention and (2) they would have wished to reduce public expenditure for the 2020 Tokyo Olympics. Further examination with a questionnaire yielded similar results for the male but not for the female sample.

**Keywords:** COVID-19; 2020 Tokyo Olympics; compact Olympics; sustainability; environment; gender equality; gender difference; Japan

## 1. Introduction

According to the sustainability plan advocated by the Tokyo Organizing Committee of the Olympic and Paralympic Games (TOC) in 2020, Japan and Tokyo, as an "advanced country/city in solving sustainability issues," would demonstrate its approach to the Sustainable Development Goals (SDGs) and enhance further sustainable development. Through an analysis of the Tokyo 2020 Games, we showcase a model addressing three main themes: climate change, resource management, and the natural environment [1]. Consistent with this purpose, the 2020 Olympics was planned to be small-scale, with minimal related public expenditure.

To reduce expenses, "Tokyo's bid promised a 'compact Olympics'; the plans envisioned 85% of event venues being concentrated within an 8-km radius of the Olympic Village" [2]. However, Olympic Games are mega events that entail enormous cost and labor for the host city and country. The average cost of the five Olympics held during 2007–2016 was $12 billion, which did not include the provision of roads, rail, airports, hotels, and other infrastructure [3]. The same was true for the 2020 Tokyo Olympics. "The cost of hosting the Games has swelled ever since Tokyo was picked to host the event. At first, the organizing committee, the Tokyo government, and the Japanese government were expecting to shell out a total of 734 billion yen (about $6.65 billion) to hold the Olympics. However, by December 2020, it had ballooned to an announced budget of 1.644 trillion yen ($14.89 billion). Add in 'Games-related expenses,' such as road improvements, and the total will likely top 3 trillion yen" [2]. Arguably, the 2020 Tokyo Olympics is considered the most expensive mega event. The International Olympic Committee (IOC) requires host cities and governments to guarantee that they will cover possible Olympics budget cost overruns.

The Olympics are expected to boost the local economy and create jobs, especially in the host city, and the Tokyo Olympics influenced the lives of not only workers in the Olympic

"OMOTENASHI" sectors but of all Japanese citizens [4]. However, various studies have found that the final costs exceed benefits for the host country [5–8], and the host country population bears a substantial portion of the cost, paying taxes over a long period [5]. Due to COVID-19, the economic returns from the 2020 Tokyo Olympics were far smaller than expected. COVID-19 exerted a significant negative economic impact on Japan because the Olympics are too commercialized, making economic loss inevitable if unexpected negative shocks occur. COVID-19 has a serious impact on the macro and micro economy [9–13], and we are now trying to adapt to the new normal era [14–16]. The 2020 Tokyo Olympics reveal that the sustainability of the Summer Olympics is vulnerable to adverse events.

As the host population, the Japanese bore enormous costs for the Olympics. To prevent the spread of COVID-19, most games were held without live spectators. The Japanese thus could not enjoy watching the games in stadiums; hence, as an entertainment event, the 2020 Tokyo Olympics ended in failure. However, commercial interests oriented the Olympics toward being a mega-event, which increased the environmental burden and accelerated global warming and climate change, leading to natural disasters. For example, many Japanese people die of heatstroke during heat waves. Almost every year, typhoons accompanied by heavy rains cause floods. In addition, several large earthquakes, such as the Great East Japan Earthquake, have had an enormous detrimental impact on the Japanese society.

This study considers the 2020 Tokyo Olympics in relation to the environment and natural disasters from the viewpoint of a sustainable society. The 2020 Tokyo Olympics slogans were "compact Olympics" and "symbol of resilience from the Great East Japan earthquake". In addition, the Olympics were expected to showcase "gender equality" and "environmental protection". Using individual-level data, this study investigates the extent to which Japanese support these slogans. The key findings are as follows: the general population supports policies that promote gender equality, disaster prevention, and environmental protection; they also wanted to reduce public expenditures for the 2020 Tokyo Olympics. After dividing the sample into male and female sub-samples and controlling for endogenous biases, these results were found to hold for the male sample only.

Various studies have assessed the economic impact of sports mega events, such as the Olympics and the FIFA World Cup [17–21]. The host city population pays costs exceeding the positive impact value of increased happiness [22]. Olympic games are broadcast on television worldwide, allowing viewers in non-host countries to enjoy the Olympics as sports entertainment without bearing the related cost, posing a free-ride problem in economic terms. Apart from the economic and entertainment perspectives, existing studies have not explored the expectations of the host country population. This study is the first to examine this issue.

The remainder of this paper is organized as follows: Section 2 describes the data and presents the methods. Section 3 presents the estimated results and interpretations. Section 4 discusses the findings, and the final section provides reflections and conclusions.

## 2. Data and Methods

### 2.1. Data

Three years after Tokyo was selected as the host city for the 2020 Summer Olympics in July 2016, data were independently collected from individuals in Japan using an internet survey. We commissioned the Nikkei Research Company to survey a representative sample of the Japanese population aged 18 to 68; Nikkei was selected owing to its reputation among Japanese researchers and experience with academic surveys. The tailor-made survey was kept open to collect a minimum of 10,000 observations. Eventually, the sample size reached 12,176 observations. The sample's demographic composition included people aged 18–67 from all parts of Japan. Next, a survey gathered a sample representative of the Japanese population. Basic characteristics, such as educational background, household income, and gender, were obtained. The questionnaire also included various specific questions about

the 2020 Tokyo Olympics and the respondents' primary school educational conditions. To address endogeneity, we added a question about the gender of the respondent's homeroom teacher in the first grade of elementary school as an instrumental variable. We asked this also in the follow-up survey conducted in 2018, in which we recruited the same respondents as had participated in 2016. Many respondents did not participate in the 2018 survey, reducing the observations to 7856. Further, some respondents did not answer the question about the gender of the teacher. This reduced the sample size to 4254. Hence, there is a possibility of selection bias. This calls for careful attention during the interpretation of the results.

Table 1 describes the variables and their mean values for the male and female samples. *COMPACT OLYMPIC, VIEW ENVIRONMENT, VIEW GENDER*, and *VIEW DISAST* are the key variables. These are discrete variables ranging from 1 (*strongly disagree*) to 5 (*strongly agree*); their mean values are around 4 on a 5-point scale. This implies that respondents prefer a compact Olympics and agree that the government should contribute to environmental protection, gender equality, and enhanced disaster prevention. Moreover, the values for women were slightly greater than for men. Therefore, women are more willing to support sustainable society policies. *FEMALE TEACHER* is a dummy variable that indicates the gender of the respondent's homeroom teacher in first grade. As explained later, *FEMALE TEACHER* is used as an instrumental variable (IV) for the two-stage IV model.

**Table 1.** Description of variables and the mean for male and female samples.

|  | Description | Male | Female |
|---|---|---|---|
| *COMPACT OLYMPIC* | *Do you agree that government should reduce public expenditure for the 2020 Tokyo Olympics?*<br>*1 (strongly disagree)–5 (strongly agree)* | 3.88 | 4.02 |
| *VIEW ENVIRONMENT* | *Do you agree that government should protect the environment?*<br>*1 (strongly disagree)–5 (strongly agree)* | 3.59 | 3.84 |
| *VIEW GENDER* | *Do you agree that government should create economic and social conditions in which women are able to fully exhibit their ability and actively participate in the workplace?*<br>*1 (strongly disagree)–5 (strongly agree)* | 3.71 | 4.01 |
| *VIEW DISAST* | *Do you agree that government should enhance disaster-prevention?*<br>*1 (strongly disagree)–5 (strongly agree)* | 4.01 | 4.27 |
| *UNIV* | *Equals 1 if respondents graduated from university, 0 otherwise* | 0.23 | 0.26 |
| *AGE* | *Age* | 43.5 | 44.1 |
| *AGE SQ* | *Squared age* | 2062 | 2204 |
| *MARRI* | *Equals 1 if respondents are married, 0 otherwise* | 0.48 | 0.58 |
| *INCOM* | *Household income* | 667 | 634 |
| *FEMALE* | *Equals 1 if respondents are women, 0 otherwise* | 0 | 1 |
| *FEMALE TEACHER* | *Equals 1 if homeroom teacher is female in the first grade in elementary school, 0 otherwise* | 0.65 | 0.76 |
| Observations |  | 2414 | 1684 |

Note: The unit of *INCOM* is 10,000 yen.

Figure 1 illustrates the distribution of *COMPACT OLYMPIC*. In line with Table 1, nearly 70% of respondents agreed or strongly agreed that the government should reduce public expenditure for the 2020 Tokyo Olympics. This indicates that most people are cautious about lavish expenditures.

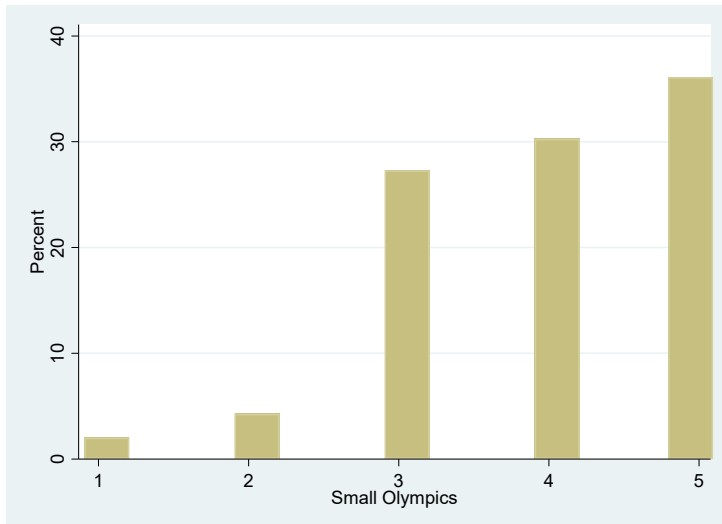

**Figure 1.** Distribution of Preference for Compact Olympics.

Figure 2 presents the geographical distribution of *COMPACT OLYMPIC*. With the exception of one prefecture (Shimane), more than half of the respondents in each prefecture preferred a compact Olympics. In addition to Tokyo, there were venues for the games in Kanagawa, Saitama, Shizuoka, Miyagi, Fukushima, and Hokkaido prefectures. In Tokyo and these areas, more than 65% of the people preferred a compact Olympics, reflecting their obligation to bear the related costs.

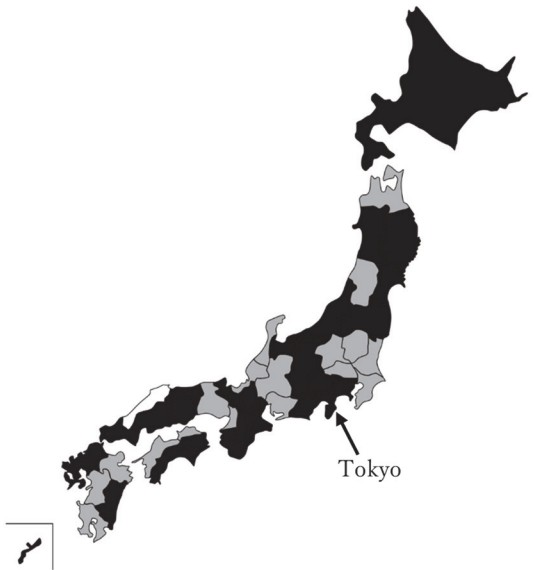

**Figure 2.** Geographical distribution of preference for compact Olympics. Note: Black shaded areas indicate prefectures where the percentage of residents who preferred small-scale Olympics is over 65%. Gray shaded areas show prefectures where the percentage is between 50% and 64%. The white area is a prefecture (Shimane) where the percentage is below 49%.

Figure 3(1)–(3) shows the distributions of *VIEW ENVIRONMENT, VIEW GENDER*, and *VIEW DISAST*. Similar to Figure 1, the majority support the argument that the government should contribute to environmental protection, gender equality, and enhanced disaster prevention. In particular, natural disasters occur frequently in Japan. In response, awareness of disaster risk management has grown [23,24]. The Japanese strongly support government engagement in disaster prevention.

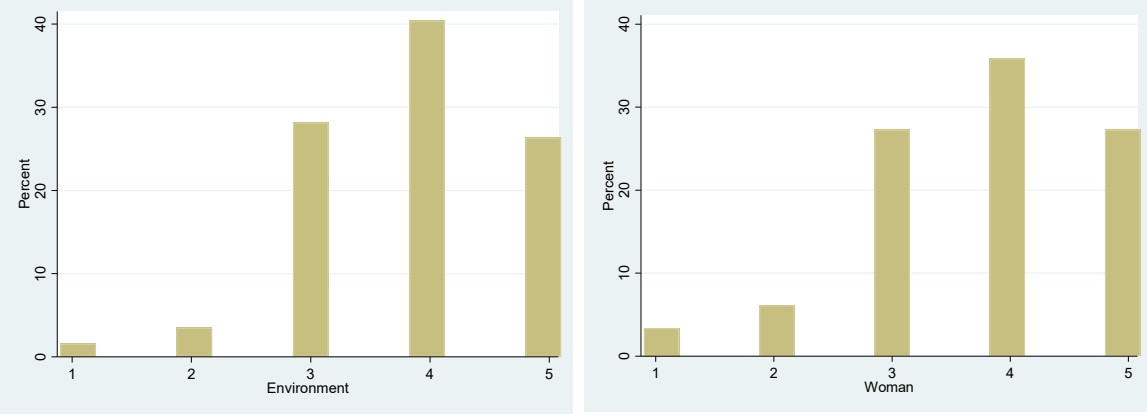

(1) Preference for environment.  (2) Preference for woman involvement.

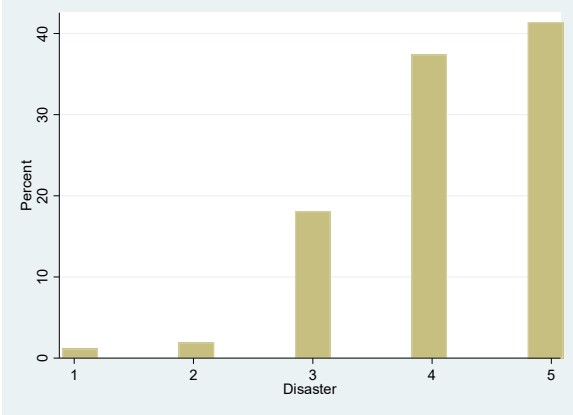

(3) Preference for disaster prevention.

**Figure 3.** Distribution of various preferences.

### 2.2. Method

Regression was applied to investigate the correlation between *COMPACT OLYMPIC* and views of government roles (*VIEW ENVIRONMENT, VIEW GENDER, VIEW DISAST*). The estimated function takes the following form:

$$COMPACT\ OLYMPIC_i = \alpha_0 + \alpha_1\ VIEW\ ENVIRONMENT\ (or\ VIEW\ GENDER,\ VIEW\ DISAST)_i + \alpha_2 UNIV_I + \alpha_3 AGE_i + \alpha_4 AGE\ SQ_i + \alpha_5\ MARRI_i + \alpha_6\ INCOM_i + \alpha_7 FEMALE_i + u_i, \tag{1}$$

where *COMPACT OLYMPIC*$_i$ represents the dependent variable for individual *i*. The regression parameters are denoted as $\alpha$, and the error term as *u*. The key independent variables are the government's role in a sustainable society. *VIEW ENVIRONMENT, VIEW GENDER,* and *VIEW DISAST* were entered separately in different estimates to examine the correlation between the view of the government's role and preference for a compact Olympics. The expected signs of the coefficients for the key variables are positive because the Olympics should be smaller if respondents were unwilling to host an Olympics, suffering from the negative effects of commercialization, such as an increase in environmental burdens. Further, respondents are thought to favor government expenditures on environmental protection and disaster prevention. Under budget constraints, it is important to consider how the government expenditure is allocated. An increase in expenditure for the Olympics, such as constructing venues, leads to a decrease in other expenditures, such as those for environmental protection, prevention of disasters, and provision of education and child rearing. In Japan, children are put on waiting lists for nursery schools, which restricts women who intend to work after having a child for whom the mother needs to care. Childcare use increases the mother's labor supply and earnings [25]. Additionally,

it improves the parenting quality of disadvantaged mothers and reduces their stress [26]. Hence, people who support women's empowerment are likely to support increases in government expenditure for child rearing by reducing expenditure for the Olympics. Similarly, people who place importance on protecting the environment and preventing disasters are more likely to support the reduction of government expenditure for the Olympics. As control variables, we included the following: *UNIV*, which is 1 if respondents graduated from university, otherwise it is 0; age (*AGE*) and its square (*AGE SQ*), the married dummy (*MARRI*) and female dummy (*FEMALE*), and household income *(INCOM)*. In Japan, there are 47 prefectures. Hence, residential prefecture dummies were used to capture differences between residential prefectures.

There is possible endogeneity bias in the baseline model because the causality between *COMPACT OLYMPIC* and the government's role is unclear. Furthermore, there seems to be a third group of factors that determine the dependent variable and key independent variables. Dependent and key independent variables can be determined simultaneously by unobserved circumstances. For example, an individual who is against commercialism might simultaneously prefer policies to reduce government expenditure and protect the environment. In the function, the error term, including the third factor, is then correlated with the key independent variables, which inevitably causes endogeneity bias. To control this, childhood experiences were used as exogenous IV to conduct a IV model estimation [27]. This study employed the same variable as IV. In the IV 2SLS (two-stage least square) model, the first-stage estimation exogenously determines the government's role.

Generally, women are more benevolent and universally concerned than men [28,29]. In Japan, after entering primary school, pupils are randomly assigned to homeroom teachers. Therefore, the sex of homeroom teachers is exogenously determined, which influences pupils' preferences. Children learn from the adults surrounding them, which shapes their worldview and social values. Preferences for trust and cooperation are transmitted through families in communities [30,31]. Men with working mothers tend to prefer working women [32], and the wives of men whose mothers worked are significantly more likely to work [33]. Hence, women influence men's views and preferences, which is called *female socialization*. Based on this argument, we assume that a female teacher's preference is transmitted to her pupils during the school experience. The homeroom teacher effect persists after pupils become adults [34,35]. Accordingly, as adults, respondents who had women teachers are expected to be more universally concerned and to view a sustainable society as more important than short-term economic benefits. As the IV variable, this study uses the female teacher dummy (*FEMALE TEACHER*), reflecting the first year of primary school. The estimated function takes the following form (2):

$$VIEW\ ENVIRONMENT\ (or\ VIEW\ GENDER,\ VIEW\ DISAST)_i \\ = \beta_0 + \beta_1\ FEMALE\ TEACHER + X'_i\ B + e_i. \tag{2}$$

Next, the key endogenous variables' predicted values are obtained for the second stage, which is equivalent to function (1). Unbiased results can thus be obtained. From the argument above, the coefficients of IV and $\beta_1$ are expected to have a positive sign in the first stage. The control variable vectors are represented by $X_i$, and $B$ is the vector of their coefficients. The set of control variables is included in the first-and second-stage functions. The estimation error of regression estimations is correlated within clusters. In the case of this study, individuals are clustered on prefectures. Hence, default standard errors can greatly overstate the estimator precision. Thus, if the number of clusters is large, statistical inference should be based on cluster-robust standard errors instead [36]. In this study, the number of clusters is 47, which is sufficiently large. In order to deal with this error, in all estimation results, robust standard errors are reported clustered on the residential prefecture.

## 3. Results

Tables 2–4 report the estimates obtained from the ordinary least squares (OLS) estimations. Tables 5–7 show the results of the IV 2SLS model. Tables 2 and 5 show the results using the whole sample. After dividing the sample into male and female sub-samples to examine gender differences, the male sample results are presented in Tables 3 and 6 and those for the female sample in Tables 4 and 7.

**Table 2.** Estimation results of the OLS model (dependent variable is *COMPACT OLYMPIC*): Sample: Male and female samples.

| | *COMPACT OLYMPIC* | | |
|---|---|---|---|
| | **(1)** | **(2)** | **(2)** |
| *VIEW* | 0.18 *** | | |
| *ENVIRONMENT* | (0.02) | | |
| *VIEW* | | 0.15 *** | |
| *GENDER* | | (0.02) | |
| *VIEW* | | | 0.25 *** |
| *DISAST* | | | (0.02) |
| *UNIV* | 0.06 | 0.06 | 0.06 |
| | (0.07) | (0.07) | (0.07) |
| *AGE* | 0.06 | 0.07 | 0.08 |
| | (0.10) | (0.10) | (0.11) |
| *AGE SQ* | 0.05 | 0.05 | 0.02 |
| | (0.10) | (0.10) | (0.10) |
| *MARRI* | −0.08 ** | −0.08 ** | −0.08 ** |
| | (0.03) | (0.04) | (0.03) |
| *INCOM* | −0.06 | −0.06 | −0.06 |
| | (0.04) | (0.04) | (0.04) |
| *FEMALE* | 0.10 *** | 0.10 *** | 0.09 *** |
| | (0.03) | (0.02) | (0.03) |
| $R^2$ | 0.06 | 0.06 | 0.08 |
| Observations | 4254 | 4254 | 4254 |

Note: Numbers within parentheses are robust standard errors clustered on the residential prefecture. Further, residential prefecture dummies are included. For convenience of interpretation, the coefficients of *AGE* and *AGE SQ* were multiplied by 10 and 1000, respectively. The coefficients of *INCOM* were multiplied by 10 and 1000. *** $p < 0.01$, and ** $p < 0.05$.

Further, Table 2 indicates a positive sign for the government's role and statistical significance at the 1% level in all columns. This is consistent with the inference presented in the previous section. For the control variables, a significant positive sign for *FEMALE* was observed in all results. Its absolute coefficient value was around 0.10, suggesting that women are more likely to support reducing Olympics expenditures by 0.10 points on the 5-point scale. This implies that women place more importance on a compact Olympics to reduce government expenditures.

Tables 3 and 4 indicate that the positive signs for the government's role are statistically significant at the 1% level for all results. The absolute values for the female sample are larger than those for the male. Support for the government's role in environmental protection, gender equality, and disaster prevention is correlated with the view of the compact Olympics. However, causality should be examined using the results in Tables 5–7.

**Table 3.** Estimation results of the OLS model (dependent variable is *COMPACT OLYMPIC*): Male sample.

| | COMPACT OLYMPIC | | |
|---|---|---|---|
| | **(1)** | **(2)** | **(2)** |
| *VIEW* | 0.17 *** | | |
| *ENVIRONMENT* | (0.03) | | |
| *VIEW* | | 0.14 *** | |
| *GENDER* | | (0.02) | |
| *VIEW* | | | 0.24 *** |
| *DISAST* | | | (0.02) |
| $R^2$ | 0.06 | 0.06 | 0.08 |
| Observations | 2517 | 2517 | 2517 |

Note: Numbers within parentheses are robust standard errors clustered on the residential prefecture. The set of control variables is equivalent to that in Table 2 although the results are not reported. *** $p < 0.01$.

**Table 4.** Estimation results of the OLS model (dependent variable is *COMPACT OLYMPIC*): Female sample.

| | COMPACT OLYMPIC | | |
|---|---|---|---|
| | **(1)** | **(2)** | **(2)** |
| *VIEW* | 0.20 *** | | |
| *ENVIRONMENT* | (0.03) | | |
| *VIEW* | | 0.16 *** | |
| *GENDER* | | (0.03) | |
| *VIEW* | | | 0.26 *** |
| *DISAST* | | | (0.03) |
| $R^2$ | 0.07 | 0.06 | 0.09 |
| Observations | 1737 | 1737 | 1737 |

Note: Numbers within parentheses are robust standard errors clustered on the residential prefecture. The set of control variables is equivalent to that in Table 2 although the results are not reported. *** $p < 0.01$.

For the IV 2SLS results, we consider the first stage of Table 5. The *F*-stat in the first stage shows the validity of *FEMALE TEACHER* as an IV variable in all columns. However, according to work of econometrics published in 2005, an F-value of 10 or higher is a strong instrument [37]. More recent study points out that the F-value must be 104.7 to clear the weak instrument problem although this view has not been established [38]. A careful attention should be called for when results are interpreted because F-stat is below 100 in all results. We only observed correlation between dependent and independent variables rather than causality in the second-stage if *F*-stat is blow 10 in the first stage.

As expected, *FEMALE TEACHER* produces a positive sign and is statistically significant at the 1% level in all columns. Therefore, respondents with a female teacher in the first grade of primary school are more likely to support government policies of environmental protection, gender equality, and disaster prevention as adults. Therefore, the IV strategy was valid. In the second stage, the signs of *VIEW ENVIRONMENT, VIEW GENDER,* and *VIEW DISAST* are positive and statistically significant at the 10% level in column and 5% level in columns (2) and (3), which is similar to the results of Table 2's OLS model. The statistical significance is lower than that in Table 2. However, its absolute value is approximately 1.50, which is significantly larger than those in Table 2 because the underestimation biases were corrected.

**Table 5.** Estimation results of the IV 2SLS model (dependent variable is *COMPACT OLYMPIC*): Sample: Male and female sample.

| | *COMPACT OLYMPIC* Second-Stage | | |
| --- | --- | --- | --- |
| | **(1)** | **(2)** | **(2)** |
| *VIEW* | 1.53 ** | | |
| *ENVIRONMENT* | (0.76) | | |
| *VIEW* | | 1.66 ** | |
| *GENDER* | | (0.72) | |
| *VIEW* | | | 1.31 *** |
| *DISAST* | | | (0.48) |
| | First-stage | | |
| *FEMALE* | 0.10 ** | 0.09 *** | 0.12 *** |
| *TEACHER* | (0.04) | (0.03) | (0.03) |
| *F*-stat. | 5.14 | 7.63 | 13.7 |
| Prob > *F* | 0.03 | 0.00 | 0.00 |
| Root MSE | 1.49 | 1.64 | 1.27 |
| Observations | 4098 | 4098 | 4098 |

Note: Numbers within parentheses are robust standard errors clustered on the residential prefecture. The set of control variables, which are included in the first and second stages, is equivalent to that of Table 2 although the results are not reported. ** $p < 0.05$ and *** $p < 0.01$.

**Table 6.** Estimation results of the IV 2SLS model (dependent variable is *COMPACT OLYMPIC*): Male sample.

| | *COMPACT OLYMPIC* Second-Stage | | |
| --- | --- | --- | --- |
| | **(1)** | **(2)** | **(2)** |
| *VIEW* | 1.48 ** | | |
| *ENVIRONMENT* | (0.67) | | |
| *VIEW* | | 1.10 *** | |
| *GENDER* | | (0.41) | |
| *VIEW* | | | 1.09 *** |
| *DISAST* | | | (0.41) |
| | First-stage | | |
| *FEMALE* | 0.10 ** | 0.14 *** | 0.14 *** |
| *TEACHER* | (0.04) | (0.04) | (0.03) |
| *F*-stat. | 5.83 | 10.2 | 13.5 |
| Prob > *F* | 0.02 | 0.00 | 0.00 |
| Root MSE | 1.53 | 1.31 | 1.21 |
| Observations | 2414 | 2414 | 2414 |

Note: Numbers within parentheses are robust standard errors clustered on the residential prefecture. The set of control variables, which are included in the first and second stages, is equivalent to that of Table 2 although the results are not reported. ** $p < 0.05$ and *** $p < 0.01$.

**Table 7.** Estimation results of the IV 2SLS model (dependent variable is *COMPACT OLYMPIC*): Female sample.

| | COMPACT OLYMPIC Second-Stage | | |
|---|---|---|---|
| | **(1)** | **(2)** | **(2)** |
| *VIEW* | 2.32 | | |
| *ENVIRONMENT* | (1.81) | | |
| *VIEW* | | 8.54 | |
| *GENDER* | | (13.6) | |
| *VIEW* | | | 2.12 * |
| *DISAST* | | | (1.06) |
| | First-stage | | |
| *FEMALE* | 0.08 | 0.02 | 0.09 ** |
| *TEACHER* | (0.06) | (0.04) | (0.04) |
| *F*-stat. | 1.96 | 0.37 | 5.23 |
| Prob > *F* | 0.16 | 0.54 | 0.03 |
| Root MSE | 1.84 | 7.09 | 1.59 |
| Observations | 1684 | 1684 | 1684 |

Note: Numbers within parentheses are robust standard errors clustered on the residential prefecture. The set of control variables, which are included in the first and second stages, is equivalent to that of Table 2 although the results are not reported. ** $p < 0.05$ and * $p < 0.1$.

As for the results using the sub-sample of males in Table 6, similar to Table 5, the *F*-stat in the first stage shows the validity of *FEMALE TEACHER* as an IV variable in all columns. However, we only observed correlation between *COMPACT OLYMPIC* and *VIEW ENVIRORNMENT* rather than causality in the second-stage in column (1) because *F*-stat is blow 10 in the first stage.

*FEMALE TEACHER* shows a positive sign and is significant at the 1% level in all columns. Therefore, the IV model was validated. In the second stage, *VIEW ENVIRORN-MENT, VIEW GENDER,* and *VIEW DISAST* show a positive sign and statistical significance in the male sample at the 1% level, and the coefficient value of each is far larger than in the results of the OLS model in Table 3. As for the results using the sub-sample of females in Table 7, the *F*-stat in the first stage does not show the validity of *FEMALE TEACHER* as an IV variable, with the exception of column (3). Further, statistical significance was only observed for *VIEW DISAST* but not for *ENVIRONMENT* or *VIEW GENDER*. Hence, among women, the view of the government's role is not associated with willingness to reduce government expenditures on the Tokyo Olympics. This is consistent with the finding of a previous study that female teachers affect male but not female pupils' preferences for corporate responsibility later in life [35]. Female teacher–male pupil matching reduces the gender difference in preferences regarding issues of the environment and gender because females tend to have a stronger interest in these issues than males do [35]. As is evident in existing studies [32,33,39–41], this study also observed cross-gender effects.

Considering Tables 2–6 together, it can be concluded that people's views about the government's role in developing a sustainable society are correlated with their viewpoint on government expenditure for the Olympics. However, IV results did not show the statistical significance only for men.

One possible explanation for the difference between the OLS and IV results using the female sample is that the third factor is correlated with dependent and key independent variables, causing estimation biases for women. For instance, women are more universally concerned than men [28,29]. This is the third factor influencing both viewpoints on government expenditures and government role. Using the randomly-assigned female teacher variable as IV, the third factor effects are controlled and a causal impact related to viewpoint

on government role is not observed. As shown in Table 1, women are more likely to prefer a compact Olympics. However, the findings of this study imply that different-gender interactions through education reduce gender preference for a compact Olympics.

## 4. Discussion

This study shows that Japanese people generally desire a sustainable society, preferring a compact 2020 Tokyo Olympics with reduced public expenditure. In opposition to Japanese public opinion, the slogan of "symbol of resilience from the Great East Japan Earthquake" was changed to represent overcoming the COVID-19 pandemic. Directly before the commencement of the Olympics, Yoshiro Mori, head of the Tokyo Olympics, said that women talk too much and that meetings with many female board directors would "take a lot of time," for which he was criticized and over which he eventually resigned [42]. During the Olympics games, contestants experienced heat waves, including high humidity. Russian tennis player Daniil Medvedev, for example, struggled with the heat during his match and told the chair umpire, "I can finish the match, but I can also die." He asked the umpire, "If I die, are you going to be responsible?" [43]. The average temperature in Tokyo has risen annually, and the climate during mid-summer does not offer a comfortable environment. The Japanese Olympic Committee could not arrange optimal conditions for the games to protect the contestants' health. A total of 130,000 meals were discarded uneaten at the Tokyo Olympics in one month [44].

Since the 1990s, the IOC has made sustainability a pillar of the so-called Olympic legacy, referring to the long-term impact the games have on both the host city and the world. This means ensuring not only that the games themselves do not harm the environment but also that they lead to permanent, positive changes for the environment [45]. This view is in line with the Japanese population's perspective. However, public expenditures for the Tokyo Olympic 2020 were far greater than in the 2016 plan. On the one hand, market mechanisms did not work because of the government's critical role in managing the Olympics. On the other hand, the IOC is essentially a sports and entertainment business, and almost 75% of its income comes from selling broadcast rights, with another 18% coming from sponsors [46]. The Olympics are thus too commercialized to truly reduce their scale, creating a gap between concept and reality for the Olympics. The scale of the Olympics has become too large to be efficiently managed due to the complexity of market-government failures, and this was especially true for the Tokyo Olympics, which did not achieve sustainable objectives because of both government failure and market failure. This stems from a type of coordination problem between players. The framework provided by the Penta helix project is useful for dealing with this issue. In this model, key stakeholders, such as NGOs, academia, and civil society, jointly participate in enhancing cost efficiency in the entire planning and implementation process, based on economies of scale and closer cooperation and exchange. The government should coordinate with these stakeholders. In comparison with these stakeholders, a broadcasting company has dominant bargaining power. Hence, the Olympics should return to an amateur model to promote sustainability.

## 5. Conclusions

The development strength of the 2020 Tokyo Olympics must be supported by all elements. The synergy between one element and the others is crucial. Therefore, the Penta helix concept or multi-stakeholder model should have been applied, where the government, academia, business entities or actors, communities, and the media [47,48] are united in coordination and committed to developing the potential of the 2020 Tokyo Olympics. Sustainability was the core idea of the Tokyo Olympics [1]. Environmental protection, empowering women, and recovery from the East Japan earthquake were key issues for the Olympics. Using individual-level data, we tested how and the extent to which Japanese people (who prefer a sustainable society) supported the compact Olympics concept advocated before the games, finding that the majority of Japanese people support the idea of

government measures to promote a sustainable society. They would like to have reduced public expenditures for the 2020 Tokyo Olympics. However, in reality, the 2020 Tokyo Olympics defied expectations and the population's demands. The IOC should organize the Olympics to reflect public opinion and contribute to realizing a sustainable society.

COVID-19 has inflicted unexpected damages on our society. When the 2020 Tokyo Olympics were held, the situation was remarkably different from that in 2016, when the survey for this study was conducted. There is a possibility that the COVID-19 pandemic caused the Japanese people to oppose hosting the 2020 Tokyo Olympics. During the era of the pandemic, it is likely that greater funding for infection control, among other measures, needs to be allocated. Such allocations do not support reducing public expenditure. We should consider this when analyzing the formation of the public view as, unfortunately, this survey was not conducted in 2021. However, these issues need to be explored in future studies.

**Funding:** This research was funded by the Japan Society for the Promotion of Science (grant number [16H03628]).

**Institutional Review Board Statement:** Ethical review and approval were waived for this study. The survey used in this study falls outside the scope of the Japanese government's Ethical Guidelines for Medical and Health Research Involving Human Subjects, and there are no national guidelines in Japan for social and behavioural research. Therefore, our study was carried out in accordance with the Ethical Principles for Sociological Research of the Japan Sociological Society, which do not require ethical reviews.

**Informed Consent Statement:** Informed consent was obtained from all subjects involved in the study. All survey participants gave their consent to participate in the anonymous online survey by Nikkei Research Company. The authors did not obtain any personal information about the participants. After being informed about the purposes of the study and their right to quit the survey, participants agreed to participate. They were provided with the option "I don't want to respond" for questions. Completion of the entire questionnaire was considered to indicate participant consent.

**Data Availability Statement:** The data presented in this study are available on request from the corresponding author.

**Acknowledgments:** We would like to thank Editage [http://www.editage.com (accessed on 5 November 2021)] for editing and reviewing this manuscript for English language.

**Conflicts of Interest:** The authors declare that there is no conflict of interest.

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
