# Peer review of "Do You Want Sustainable Olympics? Environment, Disaster, Gender, and the 2020 Tokyo Olympics"

_sustainability, doi:10.3390/su132212879_

Round 1
Reviewer 1 Report
A brief summary
This paper conducted a survey in 2016, three years after the decision was made to host the 2020 Tokyo Olympics, to find out whether people support the slogans of the Tokyo Olympics. First, it revealed that most people support the government promoting the policies of environmental protection, gender equality, and disaster prevention. They also wanted public spending on the Olympics to be kept as low as possible. The OLS results showed that the relationship between the two was positive. When the endogeneity was removed using the instrumental variable of whether or not the primary school in the first grade teacher was female, the results showed that only males did so. The contribution of this study is that it clarified the attitudes toward the burden of public expenditure for the citizens of the host country, who bear not only the benefits but also the costs of the Olympics.
Broad comments
- The relationship between the main outcome of whether or not one agrees with reducing public expenditure on the 2020 Tokyo Olympics and the dependent variable of the government promoting policies of environmental protection, gender equality, and disaster prevention was unclear. A convincing explanation of the significance of looking at the relationship between these variables is needed. In particular, there was no discussion of why there is a positive relationship between support for gender equality and support for cutting spending on the Olympics.
- The author used a survey with 12,176 observations, but the number of observations used in the analysis is much less, at 4254, about 1/3. It is necessary to write in detail what kind of sample selection was done.
- There is insufficient discussion of the impact of having a female home-room teacher in the first grade of primary school on the agreement of the policies of environmental protection, gender equality, and disaster prevention. Without a discussion of the impact of having a female home-room teacher on each of these policies, I cannot assess the validity of using this IV. In addition, since the F-values are not listed in the results, it is not possible to determine whether the IV is valid or not; unless we know whether it is a weak instrument or not, we cannot evaluate the results of the IV at this time.
- Due to COVID-19, the slogan for the 2020 Tokyo Olympics was changed to "symbol of overcoming COVID-19," but how does this relate to the results of this survey in 2016? If the same survey is conducted on the same people in 2020, when COVID-19 is prevalent, it is likely that costs for infection control, etc. will become necessary, and it is not generally considered to support a reduction in public expenditure. It is also possible that the COVID-19 epidemic may have led them to oppose holding the 2020 Tokyo Olympics. It is necessary to discuss this issue in light of these changes in the situation.
Specific comments
- Line 125 on page 4: The note for Figure 2 only describes black shaded areas, but the graph has grey and white shaded areas. It needs to be corrected.
Author Response
Author's Reply to the Review Report (Reviewer 1)
Comments and Suggestions for Authors
Thank you very much for your valuable comments. I have revised the paper in accordance with your suggestions. The revised parts are colored in red. My replies, also in red, are presented below.
Broad comments
- The relationship between the main outcome of whether or not one agrees with reducing public expenditure on the 2020 Tokyo Olympics and the dependent variable of the government promoting policies of environmental protection, gender equality, and disaster prevention was unclear. A convincing explanation of the significance of looking at the relationship between these variables is needed. In particular, there was no discussion of why there is a positive relationship between support for gender equality and support for cutting spending on the Olympics.
Reply: I have added the required explanation (lines 167–178).
- The author used a survey with 12,176 observations, but the number of observations used in the analysis is much less, at 4254, about 1/3. It is necessary to write in detail what kind of sample selection was done.
Reply: I have added the necessary explanation (lines 103–110).
- There is insufficient discussion of the impact of having a female home-room teacher in the first grade of primary school on the agreement of the policies of environmental protection, gender equality, and disaster prevention. Without a discussion of the impact of having a female home-room teacher on each of these policies, I cannot assess the validity of using this IV. In addition, since the F-values are not listed in the results, it is not possible to determine whether the IV is valid or not; unless we know whether it is a weak instrument or not, we cannot evaluate the results of the IV at this time.
Reply: I have added the necessary explanation and discussion (lines 197–203).
- Due to COVID-19, the slogan for the 2020 Tokyo Olympics was changed to "symbol of overcoming COVID-19," but how does this relate to the results of this survey in 2016? If the same survey is conducted on the same people in 2020, when COVID-19 is prevalent, it is likely that costs for infection control, etc. will become necessary, and it is not generally considered to support a reduction in public expenditure. It is also possible that the COVID-19 epidemic may have led them to oppose holding the 2020 Tokyo Olympics. It is necessary to discuss this issue in light of these changes in the situation.
Reply: I have made the addition to the conclusion section (lines 352–259).
Specific comments
- Line 125 on page 4: The note for Figure 2 only describes black shaded areas, but the graph has grey and white shaded areas. It needs to be corrected.
Reply: I have revised the note of Figure 2 to reflect your suggestion (lines 132–134).
Reviewer 2 Report
Dear Authors,
This topic is very interesting. However, the method used by the author does not give much significance, only using OLS Regression.
- The author didn't mention the level of use? Is it 1%,5%, or 10%?
- Assumptions in regression analysis need to be explained in more detail.
- In the discussion section. The author needs to add to the discussion on how to maximize this agenda by involving the Penta-helix contribution. What is the role of the government, academia, NGOs, and others?
Put this in the discussion section:
Line 52:
COVID-19 has a very serious impact on the macro and micro economy [1–5] , now we are trying to adapt in the new normal era [6–8]. In addition, COVID-19 caused a significant negative economic impact on Japan because the Olympics are too commercialized, making economic loss inevitable if unexpected negative shocks occur.
Line 299
“The development strength of the 2020 Tokyo Olympics needs to be supported by all elements. The synergy between one element with other elements is the main key. So the concept of Penta helix or multi-stakeholder where elements of the Government, academics, business entities or actors, communities or communities, and the media [9,10] are united in coordination and committed to developing the potential of the 2020 Tokyo Olympics.”
Recommended reference:
- McKibbin, W.J.; Fernando, R. The Global Macroeconomic Impacts of COVID-19: Seven Scenarios. SSRN Electron. J. 2020, doi:10.2139/ssrn.3547729.
- Djalante, R.; Lassa, J.; Nurhidayah, L.; Minh, H. Van; Mahendradhata, Y.; Ngoc, N.T. The ASEAN’s responses to COVID-19: A policy sciences analysis. PsyArXiv 2020, 368.
- Fagbemi, F. COVID-19 and sustainable development goals (SDGs): An appraisal of the emanating effects in Nigeria. Res. Glob. 2021, 3, 100047, doi:10.1016/j.resglo.2021.100047.
- Fernandes, N. Economic Effects of Coronavirus Outbreak (COVID-19) on the World Economy. SSRN Electron. J. 2020, doi:10.2139/ssrn.3557504.
- Deyshappriya, N.P.R. Economic Impacts of COVID-19 Macro and Microeconomics Evidences from Sri Lanka. SSRN Electron. J. 2020, 1–19, doi:10.2139/ssrn.3597494.
- Caraka, R.E.; Lee, Y.; Chen, R.C.; Toharudin, T.; Gio, P.U.; Kurniawan, R.; Pardamean, B. Cluster Around Latent Variable for Vulnerability Towards Natural Hazards, Non-Natural Hazards, Social Hazards in West Papua. IEEE Access 2021, 9, 1972–1986, doi:10.1109/ACCESS.2020.3038883.
- Brouder, P.; Teoh, S.; Salazar, N.B.; Mostafanezhad, M.; Pung, J.M.; Lapointe, D.; Higgins Desbiolles, F.; Haywood, M.; Hall, C.M.; Clausen, H.B. Reflections and discussions: tourism matters in the new normal post COVID-19. Tour. Geogr. 2020, 22, 735–746, doi:10.1080/14616688.2020.1770325.
- Bonacini, L.; Gallo, G.; Scicchitano, S. Working from home and income inequality: risks of a ‘new normal’ with COVID-19. J. Popul. Econ. 2021, 34, 303–360, doi:10.1007/s00148-020-00800-7.
- Caraka, R.E.; Noh, M.; Chen, R.C.; Lee, Y.; Gio, P.U.; Pardamean, B. Connecting Climate and Communicable Disease to Penta Helix Using Hierarchical Likelihood Structural Equation Modelling. Symmetry (Basel). 2021, 13, 1–21.
- Sjögren Forss, K.; Kottorp, A.; Rämgård, M. Collaborating in a penta-helix structure within a community based participatory research programme: ‘Wrestling with hierarchies and getting caught in isolated downpipes.’ Arch. Public Heal. 2021, 79, 1–13, doi:10.1186/s13690-021-00544-0.
All the best
Author Response
Ref#2 Comments and Suggestions for Authors
Thank you very much for your valuable comments. I have revised the paper in accordance with your suggestions. The revised parts are colored in red. My replies, also in red, are presented below.
English native speaker ( Editage company) edited English throuout the paper.
Dear Authors,
This topic is very interesting. However, the method used by the author does not give much significance, only using OLS Regression.
Reply: I have clarified that the results for males are robust based on the OLS and IV models. Further, I have explained that gender differences in results are consistent with the previous studies cited (lines 272–278).
- The author didn't mention the level of use? Is it 1%,5%, or 10%?
Reply: I have mentioned the percentage of significance, which can be seen in lines 253–256, and 264–269.
- Assumptions in regression analysis need to be explained in more detail.
Reply: I have added the necessary explanation (lines 197–203).
- In the discussion section. The author needs to add to the discussion on how to maximize this agenda by involving the Penta-helix contribution. What is the role of the government, academia, NGOs, and others?
Reply: I have made the changes as suggested (lines 330–337).
Put this in the discussion section:
Reply: In accordance with the referee’s suggestion, I have inserted sentences with references.
Line 52:
COVID-19 has a very serious impact on the macro and micro economy [1–5] , now we are trying to adapt in the new normal era [6–8]. In addition, COVID-19 caused a significant negative economic impact on Japan because the Olympics are too commercialized, making economic loss inevitable if unexpected negative shocks occur.
Reply: Please go through lines 53–57.
Line 299
“The development strength of the 2020 Tokyo Olympics needs to be supported by all elements. The synergy between one element with other elements is the main key. So the concept of Penta helix or multi-stakeholder where elements of the Government, academics, business entities or actors, communities or communities, and the media [9,10] are united in coordination and committed to developing the potential of the 2020 Tokyo Olympics.”
Reply: Please go through lines 342–247. Further, the recommended references have been added to the list of references, numbered “9–16” and “44–45.”
Recommended reference:
- McKibbin, W.J.; Fernando, R. The Global Macroeconomic Impacts of COVID-19: Seven Scenarios. SSRN Electron. J. 2020, doi:10.2139/ssrn.3547729.
- Djalante, R.; Lassa, J.; Nurhidayah, L.; Minh, H. Van; Mahendradhata, Y.; Ngoc, N.T. The ASEAN’s responses to COVID-19: A policy sciences analysis. PsyArXiv 2020, 368.
- Fagbemi, F. COVID-19 and sustainable development goals (SDGs): An appraisal of the emanating effects in Nigeria. Res. Glob. 2021, 3, 100047, doi:10.1016/j.resglo.2021.100047.
- Fernandes, N. Economic Effects of Coronavirus Outbreak (COVID-19) on the World Economy. SSRN Electron. J. 2020, doi:10.2139/ssrn.3557504.
- Deyshappriya, N.P.R. Economic Impacts of COVID-19 Macro and Microeconomics Evidences from Sri Lanka. SSRN Electron. J. 2020, 1–19, doi:10.2139/ssrn.3597494.
- Caraka, R.E.; Lee, Y.; Chen, R.C.; Toharudin, T.; Gio, P.U.; Kurniawan, R.; Pardamean, B. Cluster Around Latent Variable for Vulnerability Towards Natural Hazards, Non-Natural Hazards, Social Hazards in West Papua. IEEE Access 2021, 9, 1972–1986, doi:10.1109/ACCESS.2020.3038883.
- Brouder, P.; Teoh, S.; Salazar, N.B.; Mostafanezhad, M.; Pung, J.M.; Lapointe, D.; Higgins Desbiolles, F.; Haywood, M.; Hall, C.M.; Clausen, H.B. Reflections and discussions: tourism matters in the new normal post COVID-19. Tour. Geogr. 2020, 22, 735–746, doi:10.1080/14616688.2020.1770325.
- Bonacini, L.; Gallo, G.; Scicchitano, S. Working from home and income inequality: risks of a ‘new normal’ with COVID-19. J. Popul. Econ. 2021, 34, 303–360, doi:10.1007/s00148-020-00800-7.
- Caraka, R.E.; Noh, M.; Chen, R.C.; Lee, Y.; Gio, P.U.; Pardamean, B. Connecting Climate and Communicable Disease to Penta Helix Using Hierarchical Likelihood Structural Equation Modelling. Symmetry (Basel). 2021, 13, 1–21.
- Sjögren Forss, K.; Kottorp, A.; Rämgård, M. Collaborating in a penta-helix structure within a community based participatory research programme: ‘Wrestling with hierarchies and getting caught in isolated downpipes.’ Arch. Public Heal. 2021, 79, 1–13, doi:10.1186/s13690-021-00544-0.
Round 2
Reviewer 1 Report
This revised version has carefully addressed my comments in my last review. However, it is strange that all the F-values take the same value even though the first stage results such as Root MSE and coefficients of the instrument, or female teacher, are different. It is hard to imagine that all the F-values would be exactly the same when the samples are different between males and females. In particular, it seems unlikely that the F-value of the insignificant first stage in Column (2) of Table 7 would have the same F-value as the other significant estimating equations. It is necessary to check the results again. After checking the F-values again, it is necessary to add a discussion on whether there are any problems with weak instrument in the text.
Author Response
Thank you very much for your valuable comments. I have revised the paper in accordance with your suggestions. The revised parts are colored in red. My replies, also in red, are presented below.
English native spkear edited the paper. I obtained the certification of editing. However, in this web-sytem, I cannot add the certification. I send it in the next step if you request it.
Comments and Suggestions for Authors
This revised version has carefully addressed my comments in my last review. However, it is strange that all the F-values take the same value even though the first stage results such as Root MSE and coefficients of the instrument, or female teacher, are different. It is hard to imagine that all the F-values would be exactly the same when the samples are different between males and females. In particular, it seems unlikely that the F-value of the insignificant first stage in Column (2) of Table 7 would have the same F-value as the other significant estimating equations. It is necessary to check the results again. After checking the F-values again, it is necessary to add a discussion on whether there are any problems with weak instrument in the text.
Reply: The errors are due to bugs in the “Stata 15.0” program that calculated the estimation results. Specifically, in the notes to Tables 2–7 I had stated, “Numbers within parentheses are robust standard errors clustered by individual.” However, in fact I reported the robust standard errors clustered at the prefecture level because the regression model errors were correlated within clusters. Further, prefecture dummies have been included to control for the effect of residential prefectures.
F-stat cannot be appropriately obtained by using the “Stata” command for IV estimations, “ivreg,” when prefecture dummies are included and robust standard errors are clustered at the prefecture level. Hence, I used a different command, “ivreg 28,” which is available through an open source of Stata commands. By doing so, I obtained the correct F-stats. Further, in some cases there were very small differences in estimation results between the former version and the revised version due to rounding errors. However, this does not change the statistical significance and implications of the results. See the revised parts shown below;
I added some explanations regarding the calculation of robust standard errors in the case of prefecture clustering (Lines 221–226). Further, I revised Tables 2–7.
I explained the reason for including prefecture dummies (Lines: 182–184), and revised the notes below Tables 2–7.The F-stats and other results are revised accordingly in Tables 5–7.
In the main text where the IV estimations results are presented, I simply interpreted the F-stat results (Lines: 258–259, 275–276, 281–283).

Round 3
Reviewer 1 Report
I appreciate the correction of the F-values. However, the corrected F-value suggests that this instrumental variable has a weak instrument problem: Stock and Yogo (2005) state that an F-value of 10 or higher is a strong instrument, but in recent years, Lee et al. (2021) have pointed out that the F-value must be 104.7 to clear the weak instrument problem. In the second-round draft, I thought that the F-value of 990 cleared the weak instrument problem. However, in this third-round draft, I think that there is a weak instrument problem, but since most of the F-stats are below 10. This problem applies not only to estimating with the female sample, but also to estimating with the full sample and estimating for males. It is not appropriate to talk about causality from an analysis using this instrument variable. It is necessary to revise the wording to indicate causality and state the instrument has weak instrument problem.
Minor comment
F-stat in Column (2) of Table.5 is 13.7 rather than 13..7.
References
Lee, D. S., McCrary, J., Moreira, M. J., & Porter, J. R. (2021). Valid t-ratio Inference for IV (No. w29124). National Bureau of Economic Research.
Stock, J. and M. Yogo (2005) “Testing for Weak Instruments in Linear IV Regression,” In Stock, J. and Andrews, D., eds. Identification and Inference in Econometrics: A Festschrift in Honor of Thomas Rothenberg. New York: Cambridge University Press, pp. 80–108.
Author Response
Thank you very much for your valuable comments. I have revised the paper in accordance with your suggestions. The revised parts are colored in red. My replies, also in red, are presented below.
I appreciate the correction of the F-values. However, the corrected F-value suggests that this instrumental variable has a weak instrument problem: Stock and Yogo (2005) state that an F-value of 10 or higher is a strong instrument, but in recent years, Lee et al. (2021) have pointed out that the F-value must be 104.7 to clear the weak instrument problem. In the second-round draft, I thought that the F-value of 990 cleared the weak instrument problem. However, in this third-round draft, I think that there is a weak instrument problem, but since most of the F-stats are below 10. This problem applies not only to estimating with the female sample, but also to estimating with the full sample and estimating for males. It is not appropriate to talk about causality from an analysis using this instrument variable. It is necessary to revise the wording to indicate causality and state the instrument has weak instrument problem.
Reply: Following the comments, I revised the interpretation about F-stat by citing previous works suggested by the referee (Line 257-264; 282-284; 311-312). Stock and Yogo’s Criteria has been established and widely used in the field of Economics. However, Lee et al (2021) has not been published in peer referee journal. Hence, I put focus on the Stock and Yogo’s criteria and just introduce Lee et al (2021) as recent but not established one. Hence, in some of results of Tables 5 and 6, I agree that only correlation is observed based on Stock and Yogo’s criteria. At the present time, in my view, researchers have not reached agreement about the criteria of Lee et al (2021) even though the referee seems to agree it. “Sustainability” is interdisciplinary journal and so most readers cannot understand and are unlikely to be interested in the technical issue. So, I avoid touching the point in detail.
Minor comment
F-stat in Column (2) of Table.5 is 13.7 rather than 13..7.
Reply: I fixed it in Table 5.

Round 4
Reviewer 1 Report
The manuscript has been revised well. I think this manuscript is acceptable.